# Proteomics of Two Thermotolerant Isolates of *Trichoderma* under High-Temperature Stress

**DOI:** 10.3390/jof7121002

**Published:** 2021-11-24

**Authors:** Sowmya Poosapati, Prasad Durga Ravulapalli, Dinesh Kumar Viswanathaswamy, Monica Kannan

**Affiliations:** 1Department of Plant Pathology, ICAR-Indian Institute of Oilseeds Research, Rajendranagar, Hyderabad 500030, India; ravulapalliprasad@gmail.com; 2Section of Cell and Developmental Biology, University of California San Diego, La Jolla, CA 92093, USA; 3Department of Biotechnology, ICAR-Indian Institute of Oilseeds Research Rajendranagar, Hyderabad 500030, India; dineshkumarv@yahoo.com; 4Proteomics Facility, School of Life Sciences, University of Hyderabad, Gachibowli, Hyderabad 500046, India; monica_kannan2001@yahoo.co.in

**Keywords:** *Trichoderma*, thermotolerance, cell wall remodeling, MAPK, Hsf1, UPR, autophagy

## Abstract

Several species of the soil borne fungus of the genus *Trichoderma* are known to be versatile, opportunistic plant symbionts and are the most successful biocontrol agents used in today’s agriculture. To be successful in field conditions, the fungus must endure varying climatic conditions. Studies have indicated that a high atmospheric temperature coupled with low humidity is a major factor in the inconsistent performance of *Trichoderma* under field conditions. Understanding the molecular modulations associated with *Trichoderma* that persist and deliver under abiotic stress conditions will aid in exploiting the value of these organisms for such uses. In this study, a comparative proteomic analysis, using two-dimensional gel electrophoresis (2DE) and matrix-assisted laser desorption/time-of-flight (MALDI-TOF-TOF) mass spectrometry, was used to identify proteins associated with thermotolerance in two thermotolerant isolates of *Trichoderma*: *T. longibrachiatum* 673, TaDOR673 and *T. asperellum* 7316, TaDOR7316; with 32 differentially expressed proteins being identified. Sequence homology and conserved domains were used to identify these proteins and to assign a probable function to them. The thermotolerant isolate, TaDOR673, seemed to employ the stress signaling MAPK pathways and heat shock response pathways to combat the stress condition, whereas the moderately tolerant isolate, TaDOR7316, seemed to adapt to high-temperature conditions by reducing the accumulation of misfolded proteins through an unfolded protein response pathway and autophagy. In addition, there were unique, as well as common, proteins that were differentially expressed in the two isolates studied.

## 1. Introduction

Fungi belonging to the genus *Trichoderma* account for more than 60% of all registered biopesticides [1]. *Trichoderma* is used to antagonize phytopathogenic fungi, and the antifungal properties of *Trichoderma* are principally due to their ability to produce antibiotics [2] and/or hydrolytic enzymes [3] and owing to competition for space and nutrients [4]. Even if *Trichoderma* species can promote plant growth and induce resistance to biotic and abiotic stresses [5,6], their role as a bio-pesticide has primarily contributed to their commercial success as bio-agents. Nevertheless, *Trichoderma* by themselves are not immune to abiotic stresses such as moisture deficiency, higher temperature, etc., which tend to cause morphological, physiological, biochemical, and molecular changes and adversely affect the beneficial results of these bioagents. For example, soil hydrological conditions influences the growth and antagonistic properties of *Trichoderma* [7,8] and soil temperature affects the radial extension of this fungus [9].

Furthermore, agricultural practices such as solarization of the soil have been widely used to eradicate infectious pathogens from soil [10], as these practices weaken the potential of pathogens to damage crops and increase their susceptibility to bio-agents. Such practices also subject *Trichoderma,* if applied to the soil, to high-temperature stress. Therefore, there is a synergistic benefit in combining sub-lethal solarization with thermotolerant bioagents, especially *Trichoderma*, to suppress crop-damaging, temperature-tolerant pathogens in soil [11,12].

Thermotolerant strains have been identified in *Bacillus*, *Pseudomonas,* and *Beauveria bassiana* [13,14], but as *Trichoderma* is already a widely used bio-agent, the identification of thermotolerant strains of this genus would be relevant, apart from the economic significance entailed [15]. Advancements in the ‘omics’ sciences have enabled researchers to understand the mechanisms of thermotolerance in organisms such as *Saccharomyces cerevisiae*, *Metarhizium* sp. etc. [16,17,18,19], and this information has been exploited to improve the strains and stress resistance in plants [20,21,22]. In *Trichoderma,* specifically, the *hog* 1 gene (ThHog1) from the MAPK pathway was the first to be implicated in tolerance to osmotic stress [23]. Molecular studies have revealed that components of the cell wall undergo a dynamic change to adapt to stress conditions [24,25] and act as mediators of stress response [26,27,28], to activate the cell wall integrity (CWI) MAPK pathway [29,30]. Specifically, Protein kinase C (Pkc) is implicated in the maintenance of cell wall integrity (CWI) in response to different environmental stresses [31].

Fungi are known to develop different strategies, involving diverse regulatory mechanisms, to adapt to stress conditions. Heat shock response is a highly conserved pathway that results in the immediate synthesis of a pool of cytoprotective genes in the presence of diverse environmental stresses. The accumulation of misfolded proteins rapidly activates Hsf1 [18,32,33], which in turn induces several heat shock proteins that help in the proper folding of misfolded proteins during stress conditions [34,35]. The role of Hsf1 in cell wall remodeling [36] indicated a cross-talk between stress induced pathways. Only a few researchers have evaluated the role of *Trichoderma*-derived heat shock proteins in tolerance to heat, salt, and oxidative stress [20,21]. In a recent report, it was observed that elevated levels of heat shock protein from *T*. *asperellum*, TaHsp70c were specifically produced during heat and cold stress [37]. Other genes, such as glutathione transferase [38] and proteins with glucosidase activity [39], were also shown to enhance tolerance to several abiotic stresses. The studies were highly focused on understanding the effect of various stresses in pathogenic fungi of clinical importance [35,40,41,42]. Nevertheless, researchers have reviewed the role of various *Trichoderma* genes in plant stress tolerance [22].

In spite of the growing genomic resources in *Trichoderma* species (the genomes of several *Trichoderma* species have been sequenced [https://jgi.doe.gov/search-results/?q=trichoderma, accessed on 31 October 2021] and ESTs from several species of *Tricho derma* being available in the TrichoEST database [43,44], much of the research so far has focused on studies involving plant–pathogen–bioagent interactions [45], mycoparasitim [46], biocontrol related genes and enzymes [47,48], and proteases produced by bio-agents [3,49,50,51]. But there are only a few reports on the molecular changes associated with heat stress conditions in *Trichoderma* [52].

In the present investigation, we used two thermotolerant *Trichoderma* strains identified previously in our lab [15], and which differed in their level of tolerance to temperature stress. Proteins that were differentially expressed when subjected to temperature stress were identified using 2D electrophoresis and MALDI-TOF. To the best of our knowledge, this is the first report of a proteomic analysis of heat stress response in thermotolerant *Trichoderma* species.

## 2. Materials and Methods

### 2.1. Strains

The thermotolerant strains of *Trichoderma viz*., *T. longibrachiatum* 673, TaDOR673 and *T. asperellum* 7316, TaDOR7316, which had been identified previously [15], were used for this study. These strains are deposited and maintained in the microbial type culture collection (MTCC), IMTECH, Chandigarh, India. These thermotolerant strains were identified from a pool of *Trichoderma* isolates obtained from soil samples collected from various regions of India. To confirm their identity, multi-locus sequencing (Internal transcribed spacer 1 (ITS1), elongation factor1 alpha, and RNA polymerase subunit B) was performed, and the results confirmed the identity of these strains (accession numbers are provided in Appendix A).

### 2.2. Growth Conditions 

The strains were grown on potato dextrose agar at 28 °C for 7 days. Briefly, 2–3 mycelial discs from a 7-day-old culture were used to inoculate 100 mL of potato dextrose broth in 250 mL Erlenmeyer flasks and were incubated for 3 days at 28 °C and 200 rpm, followed by incubation in a static position for 4 days to allow better sporulation. Two biological replicates were used and the cultures were exposed to a temperature of 48 °C for 1 h and 4 h (treated samples). After incubation, mycelium was filtered and frozen in liquid nitrogen and stored at −80 °C until use. Samples from both strains that were not subjected to thermal stress were used as control.

### 2.3. Protein Extraction 

The protein extraction protocol described by Jacobs et al. [53] was followed, with a few modifications. In brief, approximately 1 g of fungal mycelium was ground in liquid nitrogen and suspended in 10 mL of cold (−20 °C) acetone solution, containing 13.3% Trichloroacetic acid (TCA) and 0.07% ß-mercaptoethanol. Samples were vortexed and maintained at −20 °C for at least 3 h, with intermittent shaking to allow the precipitation of protein, and then centrifuged at 14,000 rpm for 20 min at 4 °C. The pellets were washed three times with cold (−20 °C) acetone solution containing 0.07% ß-mercaptoethanol and air dried. The pellets were resuspended in labeling buffer (8 M urea, 2 M thiourea, 2% 3-[(3-cholamidopropyl)-dimethyl-ammonio]-1-propane sulfonate (CHAPS)) and 1% DTT, mixed and placed on an orbital shaker for 2 h at 37 °C to obtain complete protein solubilization. Samples were centrifuged (14,000 rpm, 60 min at 20 °C) and the supernatants were recovered. The supernatant was desalted using PD spin trap columns (GE Healthcare Bio-sciences Corp, Piscataway, NJ, USA, Code no. 28-9180-04), stored at −80 °C until use, and the protein concentration was determined by Bradford protein assay, using bovine serum albumin (BSA) as a standard.

### 2.4. Two-Dimensional Electrophoresis (2DE) and Image Analysis

Isoelectric focusing (IEF) was performed in 18 cm immobilized-pH-gradient (IPG) strips (GE Healthcare, Healthcare Bio-sciences Corp, Piscataway, NJ, USA) with a pH range of 4–7 L, rehydrated in a solution of 7 M Urea, 2 M thiourea, 4% CHAPS, 1% DTT, 2% carrier ampholytes, and 1 X Protease inhibitor cocktail (Sigma-Aldrich Inc., St. Louis, MO, USA, Catalog no. P8340). About 100 µL of total protein solution (equivalent to 500 µg) was loaded onto the focusing tray and left to be absorbed into the gel strip. The IPG strips were focused up to a total of 10 KVh using a five step program (step 500 V-for 3 h, gradient-500 V for 5 h, gradient 10 KV for 8 h until 60 KVh was reached, then finally 500 V for 10 h) in an Ettan™ IPGphor™ isoelectric focusing system (Amersham Biosciences, Chicago, IL, USA). Focused strips were equilibrated by placing them in a solution of 6 M urea, 0.05 M Tris-HCl, pH 8.8, 20% glycerol, 2% Sodium dodecyl sulphate (SDS), and 2% Dithiothreitol (DTT) for 10 min and then in 6 M urea, 0.05 M Tris-HCl, pH 8.8, 20% glycerol, 2% SDS, and 2.5% iodo-acetamide for a further 10 min. For the second dimension SDS-PAGE, the IPG strips were loaded on top of 12.5% polyacrylamide gel in an Ettan™ DALTsix Large vertical system (Amersham Biosciences, Chicago, IL, USA). Polyacrylamide gels were then electrophoresed at a constant voltage of 150 V for 60 min in Tris-glycine-SDS buffer, fixed in 40: 10% *v/v* methanol: acetic acid (overnight) and stained with sensitive colloidal Coomassie blue G-250. Gels were destained with water, until the background was clear, and stored in 40: 10% *v/v* methanol: acetic acid solution until further use. Protein patterns in the polyacrylamide gels were recorded as digitized images using a calibrated densitometric scanner (GE Healthcare, USA) and analyzed (normalization, spot matching, expression analyses, and statistics) using Image Master 2-D Platinum 6 image analysis software (GE Healthcare, USA). Spots on the gel were identified, and each spot was assigned a spot quantity (q); an approximate amount of protein based on relative spot size and intensity. Differential expression (DE) was measured as the relative ratio of q for the same spot between two comparative gels. 

For proteomic analysis, spots were analyzed using Image Master 2-D Platinum image analysis software (GE Healthcare). One-way factor ANOVA (*p* < 0.05) was performed by keeping the values of the best-matched replicate gels from the two independent experiments. The normalized volume (vol %) of each spot was automatically calculated by the Image analysis software. 

### 2.5. Mass Spectrometry and Protein Identification

In-gel digestion and matrix-assisted laser desorption/ionization time of flight mass spectrometric (MALDI-TOF MS) analysis was performed with a MALDI-TOF/TOF mass spectrometer (Bruker Autoflex III smartbeam, Bruker Daltonics, Bremen, Germany), according to the method described by Shevchenko et al. (1996), with slight modifications. Two technical replicates (for each of the two biological replicates) representing each treatment were used, and the differentially expressed spots in the comparative gels were identified and manually excised from the gels. The excised gel pieces were then destained with 100 µL of 50% acetonitrile (ACN) in 25 mM ammonium bicarbonate (NH_4_HCO_3_) at least five times. Thereafter, the gel pieces were treated with 10 mM DTT in 50 mM NH_4_HCO_3_ and incubated at 56 °C for an hour. This was followed by treatment with 55 mM iodo-acetamide in 50 mM NH_4_HCO_3_ for 45 min at room temperature in the dark (25 ± 2 °C), washed with 25 mM NH_4_HCO_3_ and ACN, dried in Speed Vac, and rehydrated in 20 µL of 25 mM NH_4_HCO_3_ solution containing 12.5 ng/µL trypsin (sequencing grade, Promega, Madison, WI, USA). This mixture was incubated on ice for 10 min and digested at 37 °C overnight. After digestion, the mixture was pulse spun for 10 s and the supernatant was collected in a fresh eppendorf tube. The gel pieces were re-extracted with 50 µL of 0.5 % trifluoroacetic acid (TFA) and ACN (1:1) for 15 min, with frequent mixing. The supernatants were then pooled together and dried using Speed Vac and were reconstituted in 5 µL of 1:1 ACN and 0.1% TFA. About 2 µL of the above sample was mixed with 2 µL of freshly prepared α-cyano-4-hydroxycinnamic acid (CHCA) matrix in 50% ACN and 0.1% TFA (1:1), and 1 µL was spotted onto the target plate. 

Proteins were identified through database searches (PMF and MS/MS) using the MASCOT program (http://www.matrixscience.com, accessed on 31 October 2021) and employing BioTools software version 3.0 (Bruker Daltonics, San Jose, CA, USA). A homology search using mass values was performed with existing digests and sequence information from the NCBInr and Swiss-Prot database, by setting the taxonomic category to ‘Other fungi’, and setting the search parameters to fixed modification of carbamidomethyl (C), variable modification of oxidation (M), enzyme trypsin, peptide charge of 1^+^, and monoisotropic. According to the MASCOT probability analysis (*p* < 0.05), only significant ‘hits’ were accepted for protein identification.

## 3. Results and Discussion

### 3.1. Morphological Differences of Thermotolerant Isolates

In the present investigation, two thermotolerant isolates of *Trichoderma*: *T. longibrachiatum* 673 (TaDOR673) and *T. asperellum* 7316 (TaDOR7316), which had previously been identified at IIOR, were used. These isolates were able to tolerate a heat shock of 52 °C for 4 h and were able to retain their morphological features after recovery from heat stress. The level of thermotolerance, however, was distinct between these isolates. The isolate TaDOR673 was highly tolerant to the heat shock condition tested, with a mean spore count (log c.f.u/mL) of 4.33 after the treatment, whereas TaDOR7316 had a mean spore count of 1.16. The two isolates exhibited distinct morphologies. TaDOR673 was a sparsely sporulating fungus with yellow pigmentation, whereas TaDOR7316 was a dense sporulating fungus (Figure 1). TaDOR673 was able to tolerate higher thermal stress, even at hyphal stage, when compared to TaDOR7316 [15].

Biochemical analysis revealed that both these isolates accumulate a higher concentration of trehalose, a known compatible solute, during heat shock [15]. We hypothesized that these morphological and physiological differences might contribute to their distinct levels of thermotolerance, and we used proteomic approaches to discern the changes associated with thermal stress in these two isolates of *Trichoderma*.

### 3.2. Protein Profiling of Thermotolerant Isolates

At temperatures higher than 48 °C, a significant reduction in protein abundance was observed (data not shown) in both the isolates, and, hence, an incubation temperature of 48 °C was selected to perform the heat stress studies, and total protein was isolated from the strains exposed to 48 °C after 1 h and 4 h. We appreciate that some transiently upregulated proteins, either within one hour of exposure or those which were altered between 1 h and 4 h, might have been missed from our analyses, as has been emphasized by Kusch et al. [40]

Using Coomassie brilliant blue staining, approximately 580 proteins were visualized in 2DE gels from each isolate (Figure 2 and Figure 3). Two replicate gels were used to construct a representative master gel (RMG) for each sample. Protein spots with significant changes in expression-level were considered to be important for thermotolerance (Figure 4 and Appendix A). A total of 32 protein spots were identified from both the isolates using MALDI-TOF-TOF (Table 1 and Table 2). Among these proteins, proteins related to heat shock response were found to be commonly affected in both the isolates. As most of the protein hits were obtained with other fungi, a search against *Trichoderma* was also carried out using BLASTP to identify the homologous proteins (Appendix A). The presence of highly conserved domains was taken into consideration using NCBI’s Conserved Domain Database (CDD). 

### 3.3. Differentially Expressed Proteins Common in the Two Trichoderma Strains 

#### 3.3.1. Proteins of Cell Wall Remodeling

There were differentially expressed proteins that were common between the two isolates tested (Table 1, Table 2, Appendix A). These proteins belonged to different cellular processes and are briefly discussed here. 

The cell wall plays a crucial role in sensing and adapting to adverse stress conditions, and earlier reports have shown that cell wall polysaccharides and lipid modifications contribute significantly to the induced heat and salt tolerance [24]. The composition of *Trichoderma* cell wall carbohydrates vary among species but are mainly composed of glucose, *N*-acetyl-glucosamine, *N*-acetylgalactosamine, galactose, and mannose [54,55]. In TaDOR673, glucose N-acetyltransferase (Spot 590), which is involved in chitin synthesis, was found to be absent after 1 h exposure, but was upregulated three-fold after 4 h, indicating the activation of chitin biosynthesis during prolonged exposure to higher temperatures. Chitin constitutes 10–30% of fungal cell walls [56] and is actively synthesized during stress conditions [25,57]. It has been hypothesized that N-acetylglucosamine (GlcNAc), a major component of fungal cell wall chitin, might be a conserved cue to morphogenesis [58] and be a mediator of cell signaling [27] during heat stress conditions.

Other cell wall proteins, such as α-1,2-mannosyltransferase (Spot 732), involved in cell wall biosynthesis were also found to be upregulated two-fold and three-fold during 1 h and 4 h exposure, respectively; indicating that N-glycosylation, a major protein modification known to occur in different cells [59], perhaps plays a crucial role in response to high-temperature stress in this organism. However, in the moderately thermotolerant *Trichoderma* strain, TaDOR7316, a reduced expression of some of the proteins involved in cell wall remodeling was observed. The enzyme UDP-galactopyranose mutase (Spot 485), which is involved in the synthesis of the cell wall component galactofuranose [60], was downregulated three-fold after 1 h exposure and was below the limit of detection at 4 h. Phospholipase B (PLB) protein (Spot 922), which is important for the turnover of phosphatidylcholine in cell membrane [61], was also repressed (~4-fold) during high-temperature stress. The physiological functions of fungal PLB are largely unknown, but recently the role of a PLB homolog was identified as a mediator of osmotic stress response in fission yeast [28]. The functional role of these proteins in imparting stress tolerance to the TaDOR7316 strain needs to be empirically determined. 

In TaDOR7316, the membranes of cellular organelles were also impacted by high-temperature stress. Phosphatidylglycerophosphatase GEP4 (Spot 901), which is involved in cardiolipin biosynthesis, was not detected during 1 h exposure to heat stress, but was expressed at a low level (compared to control) during prolonged thermal stress. Cardiolipin levels seem to be important for osmoadaptation [62] and in the cross-talk between mitochondria and vacuole [26]. Thus, in TaDOR7316, mitochondrial cell membrane signaling could be involved in the activation of other stress response pathways, which help it to ensure protection against thermal stress. 

#### 3.3.2. Proteins of Carbohydrate Metabolism

The proteins of glycolysis and gluconeogenesis pathways are highly influenced during different stress conditions, and the choice of the mechanism chosen by the organism to combat thermal stress mainly depends on the cytosol composition of the organism [63]. This probably explains why different organisms choose either gluconeogenesis [34,40,64] or glucose catabolism [42,65] to maintain cellular homeostasis during different stress conditions. 

In TaDOR673, three isoforms of enolase (Spot 329, 528, and 351) were upregulated two- and four-fold during 1 h and 4 h exposure to heat stress (Table 1); and in agreement with our results, similar observations were made in several other fungi [35,57]. With enolase being an important enzyme in carbohydrate metabolism and a major cell-envelope associated protein of *T. reesei* [66], we speculate that it could be involved in combating the heat stress conditions.

In contrast to the above, in TaDOR7316, the enzyme fructose 1, 6 bisphosphate aldolase (Spot 794), which catalyzes a reversible reaction of conversion of fructose 1,6-bisphosphate to dihydroxyacetone phosphate and glyceraldehyde 3-phosphate, was downregulated by two-fold in 1 h and three-fold in 4 h treated samples. Thus, we hypothesize that this downregulation might lead to decreased use of the TCA cycle and increased glycerol biosynthesis, as observed in osmo-regulated cells of *Aspergillus nidulans* [34]. This in turn would act as an osmoprotectant, protecting cells against dehydration or temperature stress [63].

#### 3.3.3. Proteins of the Heat Shock Response (HSR) Pathway

The role of *Trichoderma*-derived heat shock proteins in stress tolerance has been well documented [22], and it is commonly observed that under various stress conditions, the heat shock proteins Hsp70 and Hsp60 are expressed in coordination to ensure proper protein folding and maturation of nascent polypeptides [34,35].

Interestingly, in TaDOR673, Hsp70 nucleotide exchange factor Fes1 (Spot 727) and mitochondrial Hsp60 (Spot 871) were downregulated by 1.5-fold and 1.6-fold, respectively, during prolonged exposures to heat stress. These results were in agreement with earlier reports, where the authors mentioned a reduced expression of Hsp60 in fungal cells exposed to pH stress [42]. In normal cells, Fes1 specifically targets misfolded proteins recognized by Hsp70 in the ubiquitin/proteasome system (UPS), and cells lacking Fes1 are hypersensitive to induced protein misfolding and display a strong and constitutive heat shock response, mediated by heat shock transcription factor, *hsf*1 [32,33]. Thus, induced expression of heat shock transcription factor (Spot 445) in TaDOR673 (~1.5 fold) probably indicates the activation of a constitutive heat shock response in the absence of Fes1, and we hypothesize that during high-temperature stress, Hsp70 might use other dominant nucleotide exchange factors of the Hsp110 class to maintain the aggregation-prone proteins soluble in the cell, rather than targeting them to UPS for protein degradation. Moreover, it could also be possible that Hsf1 that was upregulated at higher temperatures and might be involved in the diverse cellular processes, such as cell wall remodeling, carbohydrate metabolism, energy generation, etc., during heat stress, as suggested previously [36,67].

It was interesting to observe that the small heat shock protein (sHsp), Hsp23 (Spot 1C), was induced during the initial hours of higher temperature in TaDOR7316, unlike in TaDOR673. Overexpression of Hsp23 correlated with the increased thermotolerance in *T. harzianum* reported previously [20]. In our study, overexpression of Hsp23 was confined to only during the early hours of heat stress; we speculate that there could be an increased accumulation of denatured proteins in TaDOR7316, compared to TaDOR673, during high-temperature stress. These sHsps, which are crucial for preventing stress-induced aggregation of partially denatured proteins, could rescue this organism from heat stress and aid in the restoration of the native conformations of denatured proteins when favorable conditions are restored. However, further studies are necessary to understand the role of these heat shock proteins in high-temperature stress in these isolates.

#### 3.3.4. Proteins of the Cell Signaling Pathway

A hypothetical protein (Spot 352) in TaDOR673, which possessed a conserved domain for a stress response regulator, was identified as an over-expressed protein at elevated temperatures (from 1-fold in 1 h to 10-fold in 4 h). This hypothetical protein could be the major stress signaling molecule of heat stress in TaDOR673, which in turn helps in the activation of other stress response genes. *Trichoderma* is known to use the *Hog*1-mediated mitogen-activated protein kinase (MAPK) pathway in response to stress [23] in a two-component phosphor-relay system, where a phosphoryl group is transferred from a membrane bound sensor histidine kinase to an internal receiver domain. The phosphoryl group is then shuttled through a histidine phosphor-transfer protein to a terminal response regulator (RR). When stress signals such as oxidants, high salt, etc. are detected by cells, the RR protein is dephosphorylated and is able to activate the downstream MAPK pathway to render cells adapted to the stress [29,30]. In addition, homologs of yeast RR have been identified in other fungi and were found to be involved in stress adaptation [68]. We therefore speculate that this hypothetical protein in TaDOR673 could be involved in the activation of the MAPK pathway in response to high temperature.

However, in TaDOR7316, we observed the differential expression of calcineurin, which is a serine/threonine protein phosphatase. Calcineurin is activated by Ca^2+^-calmodulin, by binding to the catalytic subunit, CnaA [69]. Interestingly in TaDOR7316 the catalytic subunit of calcineurin, which had a PP2B conserved domain (Spot C8) was not detected during high-temperature stress. However, further experiments are needed to draw conclusions about the significance of these observations for high-temperature tolerance.

#### 3.3.5. mRNA Stability

Other proteins that were common in the thermotolerant isolates were the RNA helicases. Two hypothetical proteins with conserved domains for helicases were found to be significantly downregulated in TaDOR673 (Spot 737) and TaDOR7316 (Spot 984), respectively. As is evident from Table 1 and Table 2, the expression of RNA helicases was comparatively lower in TaDOR7316 (~4–6-fold) than in TaDOR673 (~3-fold). The cellular requirement for increased RNA helicase abundance is generally associated with the alteration of the stability of the RNA secondary structure in response to stress [70,71], and our results corroborate other studies, which showed that a mutation in helicase function enhanced cellular tolerance to oxidative stress [72].

In addition to maintaining the mRNA structure, RNA helicases are also implicated in specialized processes, including cell growth, differentiation, development, small RNA (sRNA) metabolism, and response to abiotic stress [73,74,75,76]. In this context, it is interesting that downregulation of RNA helicase expression in response to stress has rarely been reported [74]. The downregulation of RNA helicases in our study points to a possible mechanism involved in induced stress tolerance when RNA helicases are either deleted/inactivated. 

In spite of the reduced levels of RNA helicases, the TaDOR7316 strain seemed to protect its mRNA pool by increased expression (~8 fold) of other mRNA protecting enzymes (Spot 1067). This hypothetical protein probably identifies and hydrolyzes the aberrantly capped mRNA produced during heat stress, in a manner similar to that observed in yeast [77].

### 3.4. Proteins Unique to TaDOR673

Apart from the major stress response pathways, the proteins involved in actin depolymerization (Spot 553) and the proteins of sulfur metabolism (Spot CN2) were downregulated during prolonged exposures to heat stress. As listed in Table 1, high-temperature stress is probably ineffective on the actin cytoskeleton during 1 h exposure, but seemed to slowly affect actin depolymerization at 4 h. Similar observations were made by Malerba et al. [78] in yeast cells exposed to mild, moderate, and high heat stress. We hypothesize that 48 °C is perhaps a moderate stress to TaDOR673 and, therefore, it probably did not undergo major cytoskeleton modifications at this level of heat stress. 

A predicted protein with a conserved domain of PWWP (Spot 817) was upregulated in 1 h (~2-fold) but downregulated 1-fold after 4 h of heat stress. PWWP domain containing proteins possess DNA or histone binding activity [79], and to our knowledge this is the first report to describe the role of a chromatin remodeling protein in *Trichoderma* under high-temperature stress. Chromatin remodeling is known to be essential for ensuring the stress-inducible binding of transcription factors, viz., Hsf1 to the majority of its targets [80], and, thus, could act as a major regulator of the stress induced response of the fungus. This would be a good candidate gene for detailed analysis, to understand the mechanisms of heat stress tolerance. 

In addition to these changes, increased expression (~7 fold) of oxidoreductases (Spot 576) was observed during prolonged heat stress conditions. As these enzymes mediate essential redox reactions in cells, they may be important in response to various stresses [64,81]. A hypothetical protein (Spot 450) was also identified as deregulated during heat stress, but its probable function could not be deciphered, due to lack of conserved domain information (Appendix A). 

### 3.5. Proteins Unique to TaDOR7316

#### 3.5.1. Unfolded Protein Response (UPR) and Protein Turnover

Among fungi, *Trichoderma* is predicted to contain a wealth of proteases (as predicted with the use of the peptidase database MEROPS http://merops.sanger.ac.uk, Rawlings et al. [50]). Of these the dominant groups include aspartyl proteases, serine proteases, subtilisin-like proteases, dipeptidyl, and tripeptidyl peptidases [82]. In the present study, two proteins belonging to the protease S8 and S53 family were upregulated at higher temperature (Spot 1095; homologous to *T. virens* subtilisin-like protease, EHK25893.1, and Spot 904; homologous to *T. reesei* tripeptidyl-peptidase 1 precursor, ETR98149.1). These proteins were upregulated ~2-fold, both at 4 h and 1 h of heat treatments. Interestingly, a hypothetical protein (Spot 920), homologous to *T. atroviride* vacuolar serine protease, ABG57252.1 was found to be downregulated during heat stress. Therefore, in TaDOR7316, upregulation of these different proteases of the UPR pathway could be essential to the targeting of the misfolded or denatured proteins for degradation. Moreover, this process helps in, both the recycling of amino acids, as well as clearing the cellular matrix [18,83].

Based on our results, we speculate that under high-temperature stress, misfolded proteins activate different enzymes of the unfolded protein response (UPR) pathway in the endoplasmic reticulum (ER), and in turn the UPR being an ER-to-nucleus signal transduction pathway might regulate a wide variety of target genes, to maintain cellular homeostasis, as reported earlier [84,85]. 

#### 3.5.2. Vacuole Biogenesis and Autophagy

We observed an increased expression of an autophagy protein, Apg6 (Spot 856), and two hypothetical proteins (Spot 1B and Spot 1026) with conserved oligomeric-Golgi domains (COG) during the stress conditions (Table 2). It was observed that under stress, when the ER-associated degradation (ERAD) pathway for misfolded proteins is saturable, cellular homeostasis is maintained by the UPR pathway, which transports the excess substrate (misfolded proteins) to the vacuole for protein turnover [85]. Studies have shown that the endocytic pathways, proteosome pathways, and regulation of autophagy are induced during heat stress in filamentous biocontrol fungi [17]. COG complex, which helps in the formation of double-membrane sequestering vesicles during autophagy [86], is principally important for retrograde trafficking within the Golgi complex and possibly for ER transport to Golgi and endosome transport to the Golgi complex [87,88,89,90,91]. 

Thus, we hypothesized that under heat stress, TaDOR7316 probably tries to maintain the native conformation of proteins during the initial hours of exposure, by inducing the expression of heat shock proteins (Hsp23 as mentioned in above sections), but during prolonged exposures to heat stress, it probably maintains cellular homeostasis by deploying the UPR pathway to transport the increased levels of denatured/misfolded proteins to vacuoles for degradation. 

Several vacuolar protein sorting complexes exist in cells, to help the efficient retrograde transport of proteins, from endosome-to-Golgi complex. Their roles in stress resistance, host cell interactions, and virulence have also been well documented [41]. In the present study, the expression of a vacuolar sorting protein with a conserved domain of VPS10 was identified in unstressed samples but was not detected under stress conditions in TaDOR7316 (Spot C7). The VPS10 encodes a Carboxypeptidase Y (CPY) sorting receptor that executes multiple rounds of sorting, by cycling between the late Golgi and a pre-vacuolar endosome-like compartment. Studies have shown that mutations in VPS10 resulted in selective mis-sorting and secretion of CPY, but had no impact on the delivery of other vacuolar proteins [92]. Thus, we speculate that in TaDOR7316, in spite of the absence of certain vacuolar sorting signal molecules, the fate of most of the misfolded proteins might be taken care of by the proteins of the UPR pathway, and that in turn might trigger vacuole biogenesis and autophagy, to maintain protein turnover and cellular homeostasis.

### 3.6. Other Proteins

A protein involved in DNA replication (spot 446, homologous to *T. reesei* QM6a DNA polymerase epsilon, XP_006964819.1) was downregulated by two-fold in 1 h and completely repressed with 4 h exposure to heat stress in TaDOR7316. Although little is known about its effects on DNA integrity and the DNA replication process, it has been shown that in yeast the absence of topoisomerase activities did not prevent the induction of either heat or radiation resistance. However, if both topoisomerase I and II activities were absent, the sensitivity of yeast to becoming thermally tolerant was markedly increased [93]. 

We identified a hypothetical protein (Spot 452) homologous to GTP cyclohydrolase of *T. reesei* QM6a, whose expression was repressed by two-fold during 1 h and was below the detection limits at 4 h of heat stress. GTP cyclohydrolase is involved in riboflavin biosynthesis, and in turn, riboflavin protects cells from stress injuries [94,95]. Elevated riboflavin is required for post-photoinductive events in sporulation of a *Trichoderma* auxotroph [96] and downregulation of this enzyme probably explains the reduced sporulation observed in TaDOR7316 during the heat stress. Similar results were also observed in *A. nidulans* and *Ashbya gossypii* exposed to oxidative stress agents [81,97]. However, further studies are required to understand the exact role of these proteins in heat stress tolerance. 

We also appreciate the fact that the differential protein expression in these two different strains of *Trichoderma* during heat stress could also be associated with the differences in biology of these two strains. Different species of *Trichoderma* are known to promote the biotic and abiotic stress tolerance of various crops using difference mechanisms, such as production of heat shock proteins, increased reactive oxygen species (ROS) production, and auxin signaling [98]. To the best of our knowledge, our study is the first to compare the biological responses of two different strains of *Trichoderma* in response to thermal stress. However, to gain further insights into the mechanism of thermotolerance in the two strains used in this study, we also carried out a microarray analysis to profile the altered expression in transcripts due to higher temperature (Poosapati et al., manuscript under preparation). By comparing the proteomic and transcriptome data, we would like to establish the correlation between the transcript and protein levels.

## 4. Conclusions

In our study, we compared the proteins profiles of two different thermotolerant isolates of *Trichoderma* under heat stress. These isolates differed in their level of thermotolerance, and, as hypothesized, we identified some putative heat stress responsive proteins that are probably responsible for the thermotolerance levels of TaDOR673 and TaDOR7316. We propose that in TaDOR673, under heat stress, the organism undergoes cell wall changes and metabolic changes, such as increased chitin production and increased enolase and trehalose biosynthesis. In addition, stress signals could be sensed by putative cell membrane sensors, which in turn transfer the signals to response regulators (RR) and activate the MAPK signaling pathway. The HSR pathway also seems to play a crucial role in the thermotolerance of TaDOR673. Based on proteomic changes, we hypothesize that Hsf1 might be involved in cell wall remodeling, to combat the heat stress in TaDOR673 strain. However, in the moderately thermotolerant isolate, TaDOR7316, heat stress probably resulted in increased accumulation of misfolded proteins, and in order to protect the cell from toxicity, misfolded proteins were targeted for degradation through UPR and autophagy. Moreover, sHsps, like Hsp23, were found to be a part of early stress responses in TaDOR7316. Thus, we observed that the superior thermotolerant isolate, TaDOR673, was able to manage the levels of heat stress through the activation of stress signaling and heat shock response pathways, whereas the moderately thermotolerant isolate, TaDOR7316 experienced severe stress during prolonged exposures to higher temperatures, and the strain could circumvent the heat stress by getting rid of toxic proteins, to maintain cellular homeostasis. Further studies are required to empirically establish the functional role of the differentially expressed proteins in these two strains during exposure to higher temperatures. Our study, thus, provides some important clues for future research in the area of *Trichoderma* proteomics.

## Figures and Tables

**Figure 1 jof-07-01002-f001:**
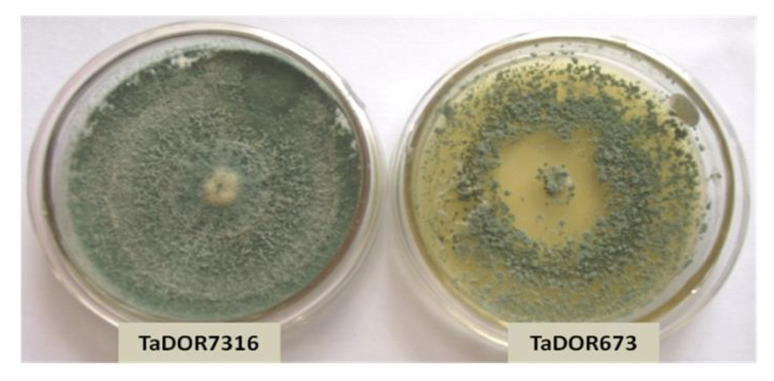
Morphology of thermotolerant isolates of *Trichoderma*.

**Figure 2 jof-07-01002-f002:**
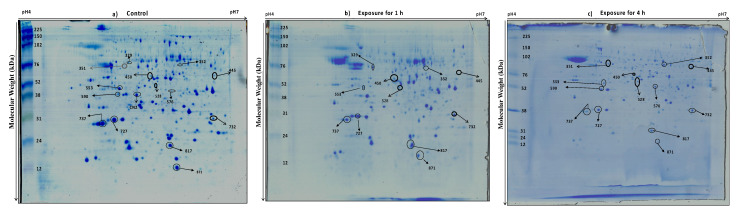
Two-dimensional gel image of the protein expression pattern in *T. longibrachiatum* 673, TaDOR673 under (**a**) control (grown at 28 °C) and heat stress conditions at 48 °C for 1 h (**b**) and 4 h (**c**). The gels were obtained in duplicates; a representative of each duplicate is shown. Identified protein spots are numbered and listed in Table 1. The pH gradient is marked above the gel, and the molecular mass protein standards (kDa) are indicated on the left of the gel.

**Figure 3 jof-07-01002-f003:**
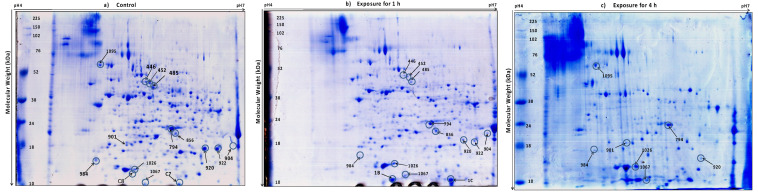
Two-dimensional gel image of the protein expression pattern in *T. asperellum* 7316, TaDOR7316 under (**a**) control (grown at 28 °C) and heat stress conditions at 48 °C for 1 h (**b**) and 4 h (**c**). The gels were obtained in duplicates; a representative of each duplicate is shown. Identified protein spots are numbered and listed in Table 2. The pH gradient is marked above the gel, and the molecular mass protein standards (kDa) are indicated on the left of the gel.

**Figure 4 jof-07-01002-f004:**
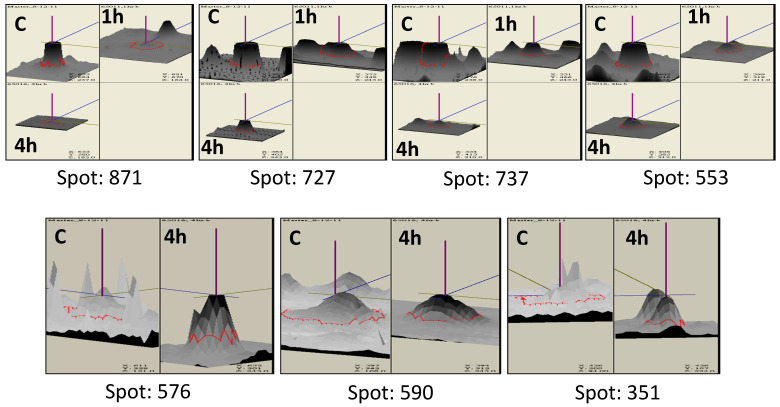
3D view of differential intensity levels of some of the protein spots identified in thermotolerant isolates of *Trichoderma* during the heat stress. Spot intensity was quantified by Image Master 2-D Platinum V6.0 image analysis software (GE Healthcare). The images show a peak for each protein spot, with a peak height that is proportional to the spot intensity. (C: control; 1 h: Heat stress at 48 °C for 1 h; 4 h: Heat stress at 48 °C for 4 h).

**Table 1 jof-07-01002-t001:** Differentially expressed proteins identified in *T. longibrachiatum* 673, TaDOR673 strain during heat stress through MALDI TOF-TOF.

Expression Pattern	Spot No ^a^	Protein Identity ^b^	Peptide Sequences Matched	Sequence ID ^c^	Score ^d^	Observed Mr (kDa) on the Gel	Theoretical Mr(kDa)/pI ^e^	% Sequence Coverage ^f^	Protein Fold Change ^g^	
									1 h	4 h
Downregulated in 1 h and 4 h compared to control	871	heat shock protein	ELKFGVEGRA KGRNVLIDQSYGSPKI RNVLIDQSYGSPKI KDGVTVAKSIVLKD	gi|294957635|	35	12	8.5/9.4	43	−4.7	−16.48
	727	Hsp70 nucleotide exchange factor	LLQQLFGGGPDEPALMR	FES1_ASHGO	12	31	32.06/5.07	5	−1.07	−1.5
	737	ATP-dependent RNA helicase DBP7	TLAYLLPIVNR	DBP7_MAGO7	20	30	89.4/9.23	1	−2.0	−3.9
	553	Protein AIM7(Altered inheritance rate of mitochondria protein 7)	MLYAGALEMIR	AIM7_YEAST	26	50	17.2/4.48	7	−4.0	−1.6
Unique to control and absent in treated samples	CN2	Probable FAD synthase	LSNIPFVFVRPR	FAD1_SCHPO	18	38	30.8/7.62	4	nd	nd
Upregulated in 1 h compared to control and 4 h	329	Enolase	GNPTVEVDVVTETGLHR	ENO_ASPFU	35	77	47.3/5.39	3	4.43	nd
Upregulated in 1 h and downregulated in 4 h compared to control	817	ISWI one complex protein 4	NEFITIFQSNNSLLLNFRILFNLR	IOC4_YEAST	22	23	55.6/5.21	5	1.95	−1.0
Upregulated in 4 h compared to control	576	Uncharacterized protein UNK4.17	YDTCIEVQADGYYLR	YEAH_SCHPO	45	45	45.8/5.31	3	nd	7.28
	590	glucose n-acetyltransferase, putative; n acetylglucosaminyltransferase,	ITLKSAPLIK	gi|241951426	52	38	56.9/8.54	2	nd	3.0
	351	Enolase	GNPTVEVDVVTETGLHR SGETEDVTIADISVGLR	ENO_EMENI	48	74	47.5/5.37	7	nd	4.05
Upregulated in 1 h and 4 h compared to control	450	predicted protein	KGSVPQPK GSVPQPKISNLSMPGVNK ISNLSMPGVNK KHNEDFEKK HNEDFEK	gi|340521168	56	65	14.5/9.36	21	2.51	2.0
	352	Stress response regulator protein 1	LTRPMVR	SRR1_LODEL	19	76	40.1/5.85	1	1.2	10.2
	445	Heat shock factor protein	SGSIQSSSDDK	HSF_YEAST	25	60	93.2/5.2	1	1.3	1.5
	528	Enolase	GNPTVEVDVVTETGLHR SGETEDVTIADISVGLR	ENO_EMENI	48	55	47.5/5.37	7	1.47	2.2
	732	GDP-Man:Man (3) GlcNAc (2)-PP-Dol alpha-1,2-mannosyltransferase	LKISPNDCENGDGFLNEMSR	ALG11_CANAL	31	31	71.2/8.68	3	2.43	3.42

^a^ Spots numbers refer to Figure 2; ^b^ Protein identity: proteins from NCBI; ^c^ Score: generated by the mascot software and represents the percentage of peptide sequence matched to the best matched protein sequence; ^d^ Sequence ID: series of digits that were assigned consecutively to each sequence record processed by NCBI; ^e^ pI: theoretical isoelectric point predicted from the amino acid sequence of the identified protein; ^f^ Sequence coverage: coverage of the amino acid sequence of the identified protein; ^g^ Fold change: the ratio of protein abundance (percentage/volume) relative to control. Negative and positive values represent reduced and increased expression of proteins, respectively; nd: not detected.

**Table 2 jof-07-01002-t002:** Differentially expressed proteins identified in *T. asperellum* 7316, TaDOR7316 proteins during heat stress through MALDI TOF-TOF.

Expression Pattern	Spot No ^a^	Protein Identity ^b^	Peptide Sequences Matched	Sequence ID ^c^	Score ^d^	Observed Mr (kDa) on the Gel	Theoretical Mr (kDa)/pI ^e^	% Sequence Coverage ^f^	Protein Fold Change ^g^	
									1 h	4 h
Unique to 1 h treated samples	1B	Conserved oligomeric Golgi complex subunit 6	ALSLPIATNIETFGR HRPHYLNSVLATFVGSR	COG6_PICGU	68	11	85.5/5.05	4	-	nd
	1C	Hypothetical protein TRIATDRAFT_306007	GPHLGQAFLPIFDVR	gi|358398918	56	10	20.6/5.61	8	-	nd
Downregulated in 1 h compared to control	485	UDP-galactopyranose mutase	YFDDCIDEALPNEDDWFTHQR CWLYFPEDDCPFYR	gi|406861020	104	41	58.9/5.64	6	−2.9	nd
	446	DNA polymerase epsilon catalytic subunit A	MMGFDSYEGGQPR VSISMIFAHPSVSGIYETR	DPOE_SCHPO	49	45	254.4/6.74	1	−2.6	nd
	922	Lysophospholi-pase NTE1	AGNPVSSLVNILNLFTSANDNVTSPSR	NTE1_CANGA	19	17	193.5/8.29	1	−4.3	nd
	452	GTP-cyclo-hydrolase II	FAVEPTWYLPGVAER SLFEHTGGSYPELITR	gi|453083792	93	45	59.1/6.23	5	−2.3	nd
Upregulated in 1 h and 4 h compared to control	1026	tRNA N6-adenosine threonylcarbamoyltransferase	MGKPLIALGLEGSANK	KAE1_SCHPO	26	12	38.06/7.55	4	2.2	4.2
	1067	mRNA cap guanine-N7 methyltransferase	SNTTMENTSGSATPKPR	MCES_ASPFU	26	11	75.2/7.59	2	8.19	7.7
Upregulated in 4 h compared to control and 1 h	1095	Kexin	STTTTSSTTTATTTSGGEGDQK	KEX2_CANAW	25	60	105.4/4.86	2	nd	2.1
Unique to control and absent in treated samples	C7	Vacuolar protein sorting/targeting protein 10	RIHLHSVTELNNVGRKIPGNTCK	VPS10_SORMK	46	10	173.2/5.63	1	nd	nd
	C8	Serine/threonine-protein phosphatase 2B catalytic subunit A1	1. TPISSAIASGSPGSPGTPTSPSIGGPPLTAWRPGHGR	PP2B1_CRYNH	20	12	72.19/5.11	5	nd	nd
Downregulated in 4 h compared to control	901	Phosphatidylglycerophosphatase GEP4	MNISGTLNTLR	GEP4_YEAST	32	18	21.1/8.87	5	nd	−1.14
Upregulated in 1 h compared to control	904	Probable tripeptidyl-peptidase SED4	ELYKMGNTFATKDPR	SED4_TRIVH	37	17	65.6/5.87	2	2.4	nd
	856	autophagy-related protein 28	DGLDEDPLSPAGSISKLKPR	gi|429854636	67	20	65.7/5.39	3	2.7	nd
Downregulated in 1 h and 4 h compared to control	920	Subtilisin-like protease 8	YIYAAQGGEGVDAYVIDTGTNIEHVDFEGR AYFSNYGK	SUB8_ARTOC	183	18	52.6/5.90	7	−4.18	−7.8
	984	Putative ATP-dependent RNA helicase C550.03c	HMIMGPSSKLISQFRLTYNMILNLLR	SKI2_SCHPO	37	15	138.6/6.30	2	−6.2	−3.7
	794	Fructose-bisphosphate aldolase, class II	EKKFAIPAINVTSSSTVVASLEAAR VNMDTDMQFAYMAGIR	gi|346326492	81	21	39.1/5.71	11	−2.1	−3.0

^a^ Spots numbers refer to Figure 3; ^b^ Protein identity: proteins from NCBI; ^c^ Score: generated by the mascot software and represents the percentage of peptide sequence matched to the best matched protein sequence; ^d^ Sequence ID: series of digits that were assigned consecutively to each sequence record processed by NCBI; ^e^ pI: theoretical isoelectric point predicted from the amino acid sequence of the identified protein; ^f^ Sequence coverage: coverage of the amino acid sequence of the identified protein; ^g^ Fold change: the ratio of protein abundance (percentage/volume) relative to control. Negative and positive values represent reduced and increased expression of proteins respectively; nd: not detected.

## Data Availability

The accession numbers of thermotolerant isolates of *Trichoderma* are available in NCBI, https://www.ncbi.nlm.nih.gov/, accessed on 31 October 2021.

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
