# Peer review of "Proteomics of Two Thermotolerant Isolates of Trichoderma under High-Temperature Stress"

_jof, 2021, doi:10.3390/jof7121002_

Round 1

Reviewer 1 Report

Major Points

- The manuscript is well written; the objective is relevant for the field since fungi from the Trichoderma genus are very important bio-agents in agriculture and are affect to abiotic factors such as high temperature.

- The methodology used although well described, is maybe the main issue concerning the quality of the work. The proteomics approach used by the authors is the classical protein separation by 2D-PAGE and identification by MALDI-TOF/TOF mass spectrometry. This approach was responsible for the establishment of proteomics as an important scientific area. However, in the last decade there was a great improvement on proteomics and the 2D-PAGE based approach was replaced (in most publications) by other methodology, specially by  LC-MS/MS bottom-up approach that permit the identification of a much higher number proteins with higher sensitivity and confidence. Also, the association of this newer approach with other quantitative tools using isobaric tags (i.e. iTRAQ, TMT) or label-free methods allowed a confident comparison of protein relative abundance between different samples. The large data generated usually require bioinformatic tools to be analyzed. On the other hand, 2D-PAGE coupled to MALDI-TOF MS is still being used by some groups. In my opinion 2D-PAGE has advantages for specific purposes such as the visualization of proteoforms, further immunodetection by western blotting, analysis of less complex samples such as immunoprecipitates etc.

- The work compares two different isolates of Trichoderma belonging to different species (T. longibrachiatum 673 and T. asperellum 7316) previously show to be thermotolerant. The first one is more tolerant to high temperatures than the other. In my opinion it would be very interesting to compare a thermotolerant and a non-thermotolerant fungus from the same species because the differences would probably point with more confidence proteins involved with the thermostolerance. Did the authors consider this possibility?

Minor Points

- The Manuscript title “Proteomic analysis reveals different sets of proteins expressed during high temperature stress in two thermotolerant isolates of Trichoderma” is true but sort of obvious since the two isolates of Trichoderma are two different species (T. longibrachiatum and T. asperellum) with different thermotolerance and phenotype. A title suggestion could be “Proteomics of two thermotolerant isolates of Trichoderma under high temperature stress”.

- Line 186. The sentence “For proteomic analysis, spots were analyzed using Image Master 2-D Platinum image….” Would fit better in the item “2.4. Two-dimensional electrophoresis (2DE) and image analysis”.

- Figure 2 –

The legend must explain the conditions of “exposure” and “ control” . In fact, I believe the “control” was not defined in the text. Was it produced at 28 oC? There was an extra incubation (1h or 4h) at this temperature?

The intensity of spots in the control gel seems to be much higher than in the treated samples. Would it be difference in staining? (I noticed that even the molecular weight markers are more intense).

- Figure 3 – the same as in Fig. 2

- Table 1 –

 The values of score for some proteins were low (for instance 12 for heat shock protein 2). How was the cut off defined?

Also, the definition of fold change should be explained specially the use of negative and positive values.

Moreover, the observed values of pI should be presented as done to Mr.

- Page 9 – Topic 3.3 refers to “Differentially expressed proteins common in the two Trichoderma strains”. However, some of the discussed proteins such as glucose N-acetyltransferase seems to be detected only in one the isolates. Did the authors meant “processes” common between the 2 strains?

- Line 300 – avoid the expression “completely absent”. Consider using  “bellow the limit of detection”

Line 319 – replace “bispohosphate” by “biphosphate”

Line 322- I did not understand why the down regulation of aldolase would increase glycerol biosynthesis since glycerol in synthesized from the products of this reaction.

Author Response

We thank the reviewer for their valuable time in reviewing our article and providing their valuable suggestions and comments to improve the quality of our article. Please find our comments below:

Major Points

- The manuscript is well written; the objective is relevant for the field since fungi from the Trichoderma genus are very important bio-agents in agriculture and are affect to abiotic factors such as high temperature.

- The methodology used although well described, is maybe the main issue concerning the quality of the work. The proteomics approach used by the authors is the classical protein separation by 2D-PAGE and identification by MALDI-TOF/TOF mass spectrometry. This approach was responsible for the establishment of proteomics as an important scientific area. However, in the last decade there was a great improvement on proteomics and the 2D-PAGE based approach was replaced (in most publications) by other methodology, specially by LC-MS/MS bottom-up approach that permit the identification of a much higher number proteins with higher sensitivity and confidence. Also, the association of this newer approach with other quantitative tools using isobaric tags (i.e. iTRAQ, TMT) or label-free methods allowed a confident comparison of protein relative abundance between different samples. The large data generated usually require bioinformatic tools to be analyzed. On the other hand, 2D-PAGE coupled to MALDI-TOF MS is still being used by some groups. In my opinion 2D-PAGE has advantages for specific purposes such as the visualization of proteoforms, further immunodetection by western blotting, analysis of less complex samples such as immunoprecipitates etc.

Author's comments: We really appreciate and understand the concern of the reviewer to use recent methodologies to have better detection of proteins. We were limited by the amount of funding provided for this project and could not use the highly sensitive methodologies which were expensive at the time of our study. With these current findings, we wish to write grant proposals and continue our research in this field to generate valuable information for the Trichoderma research community.

- The work compares two different isolates of Trichoderma belonging to different species (T. longibrachiatum 673 and T. asperellum 7316) previously show to be thermotolerant. The first one is more tolerant to high temperatures than the other. In my opinion it would be very interesting to compare a thermotolerant and a non-thermotolerant fungus from the same species because the differences would probably point with more confidence proteins involved with the thermotolerance. Did the authors consider this possibility?

Author's comments: We really appreciate the reviewer’s idea of comparing the thermotolerant and a non-thermotolerant strain of Trichoderma. The thermotolerant isolates identified in our study were able to survive at temperatures optimum for other Trichoderma species and were also able to tolerate high temperature stress conditions. Hence, we compared the heat stressed samples with the untreated to compare the changes in protein expression. At that point of time, we just were curious to know what are the genes or proteins that make them thermotolerant and did not focus on comparing the non-thermotolerant with the thermotolerant isolates.

We compared two thermotolerant strains in our study because from our previous experiments we learned that these two isolates show different morphological and biochemical changes in response to heat stress hence we got curious to understand the biology of these isolates. These thermotolerant isolates were identified from soil samples collected from different regions in India. We appreciate the inherent differences associated with different strains of Trichoderma and in order to have a better comparison of heat stress tolerance, we might have to find similar strains of Trichoderma. In order to focus on the major goals of the project, we had to do this experiment to generate potential information about the proteins involved in high temperature stress tolerance in thermotolerant Trichoderma isolates.

Minor Points

- The Manuscript title “Proteomic analysis reveals different sets of proteins expressed during high temperature stress in two thermotolerant isolates of Trichoderma” is true but sort of obvious since the two isolates of Trichoderma are two different species (T. longibrachiatum and T. asperellum) with different thermotolerance and phenotype. A title suggestion could be “Proteomics of two thermotolerant isolates of Trichoderma under high temperature stress”.

Author’s comments: We appreciate the reviewer’s suggestion and changed the title.

- Line 186. The sentence “For proteomic analysis, spots were analyzed using Image Master 2-D Platinum image….” Would fit better in the item “2.4. Two-dimensional electrophoresis (2DE) and image analysis”.

Author’s comments: Included the sentence in section 2.4 as suggested by the reviewer

- Figure 2 –

The legend must explain the conditions of “exposure” and “control”. In fact, I believe the “control” was not defined in the text. Was it produced at 28°C? There was an extra incubation (1h or 4h) at this temperature?

The intensity of spots in the control gel seems to be much higher than in the treated samples. Would it be difference in staining? (I noticed that even the molecular weight markers are more intense).

- Figure 3 – the same as in Fig. 2

Author’s comments: Included the details in the legends of figure 2 and 3 as suggested by the reviewer’s comments. We had loaded equal amount of protein for the control and treated samples. We have observed reduced expression of proteins in treated samples after exposure to heat stress for both the isolates of Trichoderma. We believe this could be due to the thermal stress conditions used in our study. We also do not rule out the possibility of difference in handling errors while staining the gels. To avoid these practical errors, we had repeated the experiment for at least three times and used 2 gels that showed consistent protein expression to select differentially expressed proteins across the control and treated samples.

- Table 1 –

 The values of score for some proteins were low (for instance 12 for heat shock protein 2). How was the cut off defined?

Author's comments: Score is generated by the mascot software when we use our peptide sequence as query to find the best match for our peptide sequence from NCBI database. This value represent the percentage of sequence matched to the best matched protein sequence. The same description is included in the legends of Table 1 and 2.

Also, the definition of fold change should be explained specially the use of negative and positive values.

Author's comments: Details included in the legends of Table 1 and 2.

Moreover, the observed values of pI should be presented as done to Mr.

Author's comments : We have tried to include the exact information generated by Mascot search engine in a table format that is easy for the readers. Authors did not understand what is the exact change requested by the reviewers. Please let us know the details so that we can include the necessary changes in the manuscript.

- Page 9 – Topic 3.3 refers to “Differentially expressed proteins common in the two Trichoderma strains”. However, some of the discussed proteins such as glucose N-acetyltransferase seems to be detected only in one the isolates. Did the authors mean “processes” common between the 2 strains?

Author's comments: Yes, as the reviewer correctly pointed out, our idea of grouping these differentially expressed proteins into one category is based on the common processes they are involved in. Hence although this protein was present in TaDOR673, we grouped it into the category of “proteins of cell wall remodelling” because there were proteins expressed in TaDOR7316 that were also part of similar processes.

- Line 300 – avoid the expression “completely absent”. Consider using “below the limit of detection”-

Author's comments: Repharsed the sentence “completely absent” with phrases like “below the limit of detection” or “not detected” wherever applicable

Line 319 – replace “bispohosphate” by “bisphosphate”-

Author's comments: Changes included as applicable

Line 322- I did not understand why the down regulation of aldolase would increase glycerol biosynthesis since glycerol in synthesized from the products of this reaction.

Author's comments: We observed in our study that in TaDOR7316, fructose 1, 6- bisphosphate aldolase is downregulated in response to heat stress. Aldolase is involved in the reversible conversion of fructose 1,6-bisphosphate to dihydroxyacetone phosphate and glyceraldehyde 3-phosphate. Hence its reduced production would result in reduced pyruvate levels, which in turn would reduce the cells’ utility of TCA cycle, resulting in reduced NADH generation. However, with continuous uptake of glucose and production of fructose 1,6- bisphosphate, the cells might change the flux towards glycerol production which is a protectant of abiotic stresses (Kim et al., 2007). In our study, we discussed very few significant proteins but hence could have missed some of the other proteins involved in this pathway to further defend our hypothesis. Our study is supported by the similar observations made by Kim et al. 2007 in Aspergillus nidulans in response to osmotic stress.

Author's once again thank the reviewers and the editor for the critical review of our article. 

Reviewer 2 Report

In this manuscript, the authors investigated two thermotolerant isolates TaDOR673 and TaDOR7316 by using two-dimensional gel electrophoresis and MALDI-TOF-TOF. 32 differentially expressed proteins were identified. Sequence homology and conserved domains analysis revealed that the thermotolerant isolate TaDOR673 seemed to employ the stress signaling MAPK pathways and heat shock response pathways to combat the stress condition whereas the moderately tolerant isolate TaDOR7316 seemed to adapt to high temperature conditions by reducing the accumulation of mis-folded proteins through unfolded protein response pathway and autophagy. The results are interesting, but the evidences of some conclusions need to be strengthen.

  1. To confirm the present findings and support the conclusions, the authors should analyze two isolates by microarray analysis. Indeed, these author have done this analysis (Poosapati et al. Manuscript under preparation). I suggest they put microarray data in this manuscript.
  2. In this manuscript, two Trichoderma strains, TaDOR673 and TaDOR7316, were used. They are different species and displayed different thermotorlerances. It is not surprised to find these two isolates adopt different strategy to deal with heat shock. Therefore, the authors should add some discussion about this issue.

Author Response

We thank the reviewer for their valuable time in reviewing our article and providing their valuable suggestions and comments to improve the quality of our article.

In this manuscript, the authors investigated two thermotolerant isolates TaDOR673 and TaDOR7316 by using two-dimensional gel electrophoresis and MALDI-TOF-TOF. 32 differentially expressed proteins were identified. Sequence homology and conserved domains analysis revealed that the thermotolerant isolate TaDOR673 seemed to employ the stress signaling MAPK pathways and heat shock response pathways to combat the stress condition whereas the moderately tolerant isolate TaDOR7316 seemed to adapt to high temperature conditions by reducing the accumulation of mis-folded proteins through unfolded protein response pathway and autophagy. The results are interesting, but the evidences of some conclusions need to be strengthened.

  1. To confirm the present findings and support the conclusions, the authors should analyze two isolates by microarray analysis. Indeed, these authors have done this analysis (Poosapati et al. Manuscript under preparation). I suggest they put microarray data in this manuscript.

Author's comments: We appreciate the comments from the authors. As the authors have suggested, we have conducted the microarray experiments and have the list of differentially expressed genes in the thermotolerant isolates of Trichoderma. We have outsourced our microarray work to Genotypic Technology, Bangalore, India. The data is huge and we planned to publish it as an independent publication to focus specifically on the differential regulation of genes in response to heat stress. We fear that including the microarray data might dilute the focus on proteomics results detailed in this article and we fear that we might have to rewrite the entire manuscript. We can provide a list of differentially regulated genes as a supplementary table if possible. We request the reviewer to kindly reconsider our genuine practical difficulties in addressing this suggestion.

  1. In this manuscript, two Trichoderma strains, TaDOR673 and TaDOR7316, were used. They are different species and displayed different thermotolerances. It is not surprised to find these two isolates adopt different strategy to deal with heat shock. Therefore, the authors should add some discussion about this issue.

Author’s comments: We really want to thank the reviewer for bringing up this point. We do appreciate that the differential responses of different strains of Trichoderma could be inherent to the biology of these strains. We have included these suggestions in our “Discussion section” in lines 493-500.

We once again thank the reviewer and the editor for their valuable time and effort in reviewing our manuscript. We would really appreciate if our honest concerns are taken into consideration for accepting our manuscript.

Reviewer 3 Report

As I read this manuscript, this work is good and realy big efforts, and this work in my exiperiance for More than 30 years, and this work good written and  acceptable at its present form I have not any Suggestion for any improvement, this work is hard work.

Author Response

Comments and Suggestions for Authors

As I read this manuscript, this work is good and really big efforts, and this work in my experience for More than 30 years, and this work good written and acceptable at its present form. I have not any Suggestion for any improvement, this work is hard work.

Author's comments: We are thankful to the reviewer for accepting our manuscript and recognizing the importance of our work.

Round 2

Reviewer 1 Report

The authors addressed the comments/suggestions. In the future they intend to revisit this model using a LC-MS/MS based proteomics approach that will provide a larger number of identifications.

There are two minor points they should check

  • In response to my question about Mascot score they wrote in the legends of Tables 1 ad 2 : “Score: It is generated by the mascot software and it represents the percentage of peptide sequence matched to the best matched protein sequence.” I am not sure it is correct because a protein have a score higher than 100 ( UDP-galactopyranose mutase- score 104).
  • Line 504 –the word “mechanisms” is misspelled.

Author Response

Reviewer 1:

We greatly appreciate the reviewer’s response to our comments. We addressed the additional comments/suggestions requested by the reviewer below:

The authors addressed the comments/suggestions. In the future they intend to revisit this model using a LC-MS/MS based proteomics approach that will provide a larger number of identifications.

Authors are really grateful to the reviewer for understanding our concerns and reconsidering our request.

There are two minor points they should check

  • In response to my question about Mascot score they wrote in the legends of Tables 1 ad 2 : “Score: It is generated by the mascot software and it represents the percentage of peptide sequence matched to the best matched protein sequence.” I am not sure it is correct because a protein have a score higher than 100 (UDP-galactopyranose mutase- score 104).

Authors comments: For some of the proteins, there were two or more peptide sequences that matched a portion of the single protein sequence. The mascot software usually generates a score based on the query sequence/sequences that best match to a protein and represent the value that is additive of all the scores for each matched peptide sequence used in the query together. We have shown below the mascot result obtained for sample 485 for reference.

MALDI TOF/TOF Report

Sample ID: 485

1.    

gi|406861020    Mass: 58954    Score: 104    Matches: 2(1) Sequences: 2(1)

UDP-galactopyranose mutase [Marssonina brunnea f. sp. 'multigermtubi' MB_m1]

Check to include this hit in error tolerant search

Query  

Observed  

Mr(expt)  

Mr(calc)  

 ppm  

Miss 

Score 

Expect 

Rank 

Unique 

 Peptide

3  

1968.8643  

1967.8570  

1966.8018  

536 

0  

34  

0.71 

1  

U   

 K.CWLYFPEDDCPFYR.A

4  

2686.9336  

2685.9263  

2685.1078  

305 

0  

69  

0.00018 

1  

U   

 K.YFDDCIDEALPNEDDWFTHQR.I

Protein View: gi|406861020

UDP-galactopyranose mutase [Marssonina brunnea f. sp. 'multigermtubi' MB_m1]

Database:

NCBInr

Score:

104

Nominal mass (Mr):

58954

Calculated pI:

5.64

Taxonomy:

Marssonina brunnea f. sp. 'multigermtubi' MB_m1

Sequence similarity is available as an NCBI BLAST search of gi|406861020 against nr.

Search parameters

MS data file:

Enzyme:

Trypsin: cuts C-term side of KR unless next residue is P.

Fixed modifications:

Carbamidomethyl (C)

Variable modifications:

Oxidation (M)

Protein sequence coverage: 6%

Matched peptides shown in bold red.

1

MADINVDILV

IGAGPTGLGA

AKRLNQIDGP

SWMIIDSNEK

AGGLASTDVT

51

PEGFLYDVGG

HVIFSHYKYF

DDCIDEALPN

EDDWFTHQRI

SYVRCKEQWV

101

PYPFQNNISM

LPTEEQVKCV

DGLIDAALAA

RTATDKPKDF

DEWIVRNIGV

151

GIADLFMRPY

NFKVWAVPTT

KMQCAWLGER

VAAPDVKTVV

KNIMLNKTAG

201

NWGPNATFRF

PARDGTGGIW

IAVAKTIPNE

KKLFGEQGEV

SKVNADAHTV

251

TLANGKTIGY

KKLITTMAVD

QLVEQMEDKE

LISLSKGLFY

SSTHVIGVGL

301

RGARPERIGD

KCWLYFPEDD

CPFYRATIFS

NYSPYNQPAD

SVKLPTQYLA

351

DGSKPESSEP

KEGPYWSIML

EVSESSMKPV

DRANILKDTI

QGLINTGMLK

401

SDDEIISTYH

RRFDHGYPTP

SLERDGVLKE

LLPKLQAMDI

YSRGRFGSWK

451

YEVGNQDQSF

MLGVEAVDHV

VHGAVELTLN

YPDLVNGRKN

DERRLGSTAL

501

LKRNAEPTKS

NSAREIPTRE

RASSKA

Bottom of Form

Search Parameters

Type of search         : MS/MS Ion Search

Enzyme                 : Trypsin

Fixed modifications    : Carbamidomethyl (C)

Variable modifications : Oxidation (M)

Mass values            : Monoisotopic

Protein Mass           : Unrestricted

Peptide Mass Tolerance : ± 536.5 ppm

Fragment Mass Tolerance: ± 1.95 Da

Max Missed Cleavages   : 0

Instrument type        : MALDI-TOF-TOF

Number of queries      : 4

Line 504 –the word “mechanisms” is misspelled.

Corrections were made as suggested by the reviewer. Due to the changes already made in the manuscript, we believe the line number changed from 504 to 514 so the changes made can be seen in line 514 of revised version of the manuscript.

Reviewer 2 Report

The authors have improved the manuscript.

Author Response

Reviewer 2: Comments and Suggestions for Authors

The authors have improved the manuscript.

We greatly appreciate the reviewer’s time for revisiting our manuscript and accepting our corrections made to the manuscript. We thank the reviewer for all the valuable suggestions and comments requested for the improvement of our manuscript.
